# High prevalence of active trachoma and associated factors among school-aged children in Southwest Ethiopia

**Dawit Getachew**[1]*, **Fekede Woldekidan**[2], **Gizachew Ayele**[3], **Yordanos Bekele**[1],
**Samrawit Sleshi**[1], **Eyob Tekalgn**[3], **Teshale Worku**[3], **Mengistu Ayenew**[1], **Biruk Bogale**[1],
**Abyot Asres**[1]

**1** Department of Public Health, School of Public Health, College of Medicine and Health Sciences, Mizan Tepi University, Mizan Aman, Ethiopia, **2** Department of Public Health, Ethiopian Defense University, Bishoftu, Ethiopia, **3** Department of Microbiology, College of Medicine and Health Sciences, Mizan-Tepi University, Mizan Aman, Ethiopia

* getdawit2011@gmail.com

## Abstract

### Background

Active trachoma is a highly contagious ongoing stage of trachoma that predominantly occurs during childhood in an endemic area. This study assessed the prevalence and factors associated with active trachoma among school-aged children.

### Methodology/Principal findings

A community-based analytical cross-sectional study was done from March 1st to June 30th, 2021, in Southwest Ethiopia's people's regional state. A total of 1292 school-aged children were surveyed. The quantitative data were collected using a pre-tested, structured interview-based questionnaire and observation check list. The World health organization (WHO) simplified trachoma grading system was used to assess stages of trachoma. In this study, the prevalence of active trachoma was 570(44.1%), 95% CI (41.4, 46.9). Also, age group 6–10; being female; flies at household (HH), flies on child's face, improved water source, improved sanitation, presence of ocular discharge, presence of nasal discharge, and unclean faces of the child were significantly associated with active trachoma.

### Conclusions/Significance

The very high prevalence of active trachoma in the study area is significantly associated with; age group 6–10, female gender, presence of flies in household and on child's face, presence of ocular and nasal discharge, unclean faces, improved water source, improved sanitation in the household. Thus, environmental sanitation and facial cleans trachoma elimination strategy should be intensified in the study area.

**Data Availability Statement:** The data is included with the submission as a supporting file.

**Funding:** Mizan-Tepi University (https://mtu.mizantepiuniversity.net/) funded this research

under the broader research theme 'prevalence of selected neglected tropical diseases among school-age children in Southwest Ethiopia's regional state' by the reference number MTU/59/136/44/21 on February 2, 2021. The grant recipients were DG and FW. Mizan-Tepi University announced a call for the proposal, provided the research theme, evaluated the proposal, and supervised the overall research project management. The funders had no role in study design, data collection and analysis, the decision to publish, or the preparation of the manuscript.

**Competing interests:** The authors have declared that no competing interests exist.

## Author summary

Trachoma is the leading cause of preventable blindness, due to repeated bacterial infection of the eye. The early and contagious stage of the disease, known as active trachoma, predominantly affects children. The impact of the disease includes permanent visual impairment, dependency and stigma.

The WHO targeted to eliminate trachoma by 2030 as a public health concern in each endemic country. Through the implementation of the SAFE strategy (surgery to treat the blinding stage; antibiotics to clear infection; environmental improvement; improving access to water and sanitation).

In Ethiopia majority of the districts are still trachoma endemic. Thus, this study assessed the prevalence and factors associated with active trachoma among school-aged children in Southwest Ethiopian Peoples Regional State. The region is highly endemic for several neglected tropical diseases including trachoma.

In this study 44.1% of the participant had active trachoma. Also, age groups 6–10, being female, improved water sources, and sanitation, presence of flies in the house and on the child's face, ocular and nasal discharge, unclean faces were factors associated with the occurrence of active trachoma.

Intensive scaling up of the facial cleanness and environmental improvement components of the SAFE strategy helps reduce the high prevalence of active trachoma.

## Introduction

Trachoma is a neglected tropical disease caused by the bacterium Chlamydia trachomatis [1]. The disease is transmitted by direct personal contact and by flies that have come into contact with discharge from an infected person's eyes or nose [2]. Trachoma limits the education and economic empowerment of infected individuals, which results in dependency and stigma [3]. Communities who lack access to water, sanitation, and hygiene (WASH) were primarily affected by trachoma [4,5].

Trachoma has five clinical stages: trachoma follicular, trachoma intense, trachoma scarring, trachomatous trichiasis, and corneal opacity [1,6]. Active trachoma, an early stage of trachoma, is conjunctival inflammation, which can lead to, scarring, and blindness if untreated [3,6–10]. Globally, from 166.6 million people lived in trachoma-endemic areas in 2019, 87% were from African and 46% from Ethiopia [11–15].

A multitude of factors were associated with active trachoma, including age of the child and the number of children [16–18]; as access WASH at HH [2,12,17–19]; presence of ocular and nasal discharge, habit of face washing, not using soap while washing the face, and sharing towels [12,16,18].

The SAFE strategy is being implemented in endemic countries eliminate trachoma as a public health importance [20–23]. But, in Ethiopia 80% of districts were endemic and majority of HH lack access to improved WASH [13,24–26]. However, no research has assessed the prevalence of active trachoma in the study area. This study aimed to assess the prevalence of active trachoma and associated factors among school-aged children in Southwest Ethiopia Peoples Regional State (SWEPRS).

## Methods

### Ethics statement

The Ethics and Research Committee of Mizan-Tepi University approved and issued the ethical clearance. Also, permission was obtained from the zonal health department. Informed written consent was sought from parents and assent from children aged 7–16 years. All participants were informed the aims, purpose, risks, and benefits of the study. Throughout the study, confidentiality, anonymity, and the freedom to withdraw from the study at any time were respected. The data was kept safe under strict supervision by the principal investigator. Children who have active trachoma were treated by tetracycline eye ointment, and referral linkage was made with a health facility.

### Study design and settings

A Community based Analytical cross- sectional study was done in SWEPRS from the 1st of March to the 30th of June, 2021, as part of a broader research theme 'prevalence of selected neglected tropical diseases among school-aged children in SWEPRS.' The region is divided into six zones: Kaffa, Bench-Sheko, Sheka, West-omo, Dawro, and Konta Zones. In these zones, there are 57 districts (41 rural districts and 16 city administrations). There are also kebeles in the region (kebele is the lowest legal administrative division in Ethiopia). The total population of the region is 3,368,385 [27]. In the region, there are one teaching university hospital, two general hospitals, 10 primary hospitals, 134 health centers, and 836 health posts.

### Study population and eligibility criteria

All school-aged children whose age was found in the range of 6–16 year in SWEPRS were the source population. All school-aged children whose age was found in the range of 6–16 year and lived in selected HH during the data collection were the study population. All school-aged children, irrespective of their school enrollment status, paired with HH heads who lived in the selected kebele for at least 6 months were included in the study. Children who were severely ill and unable to respond or participate were excluded from the study.

### Sample size determination and sampling technique

The sample size was calculated using Epi info software, For the first objective, with the assumptions: prevalence of active trachoma 17.5% [13], margin of error 5%, Z/2 95% CL = 1.96, and design effect 2. The calculated sample was 444. For the second objective; considering the assumption: Z/2 95% CL = 1.96, the open defecation-free status variable with OR 2.52, percent of outcome in exposed 85.5% [13], the calculated sample size was 1200 and adding 10% for non-response rate makes the final sample size 1320.

A multi-stage sampling technique was used. First, three zones: Bench-Sheko, Kaffa, and West-Omo were selected randomly. The selected zones were stratified in to rural districts and city administrations to select 12 rural districts and 4 city administrations. At least 30% kebeles in each selected strata were selected by using the lottery method. The sample was proportionally allocated to each selected kebele based on their contribution. Finally, a sampling frame was prepared by listing HH that have eligible individual, and participants were selected using a simple random sampling technique. If more than one school-aged child lived in the same HH, one child was selected by lottery method.

### Variables of the study

**Dependent variable.** Presence or absence of active trachoma.

**Independent variable.** Socio-demographic characteristics include: the sex of the HH head, education status of the HH head, occupation of HH the head, family size and number children aged 1 to 9 years in the HH. **Behavioral factors include**: frequency of face washing, using soap, discharge from the eye and nose, presence of flies on child face.

Environmental and WASH related factors includes: Prescence of feces around the HH, waste disposal, source of water, distance to water source and availability of latrine.

## Operational definitions

**Active trachoma**: the presence of five or more follicles greater than 0.5 mm in diameter in the central part of upper tarsal conjunctiva or inflammatory thickening of the tarsal conjunctiva that obscures more than half of the normal deep tarsal vessels in either of the child's eyes [9,28].

**Improved water sources**: adequately protected from outside contamination, in particular from fecal matter HH connections, public standpipes, boreholes, protected dug wells, protected springs and rainwater collection [29,30].

**Improved sanitation**; is flush or pour-flush to piped sewer system, septic tank pit latrines, ventilated-improved pit latrines, or pit latrines with slab or composting toilets [29,30].

**Unclean face: A**ny dust and food on the face during clinical examination [16].

## Data collection methods and materials

The data were collected using a structured interviewer-administered questionnaire, and observational checklist prepared and adopted after reviewing relevant literature [16,18], and a simplified WHO trachoma grading system [9,28]. The interview questionnaire measures sociodemographic characteristics, and behavioral factors. The interview was conducted by four trained public health professionals in the participants compound where they are comfortable. The observation check list was used to measure the HH and surrounding for WASH status. The observation was done by four trained environmental health professionals immediately after the interview was completed. The clinical trachoma grading was done by four trained BSc nurses to clinically diagnose each eye of the child.

## Data quality control

The questionnaire was prepared in English, then translated into Amharic and retranslated back to English to keep consistency. Also, the interview questionnaire was pretested on 5% of the sample in area where the study was not done. In addition, the data collectors and supervisors were trained on the purpose of the study, data collection technique for two days. Supervisors and investigators checked the collected data for completeness, accuracy.

## Data processing and analysis

The data were coded, cleaned, and entered into Epi Data Version 4.02, then exported to SPSS version 23 for further analysis. In bivariable logistic regression analysis variables with p-value less than 0.2 were entered in to multiple logistic regression. In multiple logistic regression variables with p-value less than 0.05 were considered as significantly associated with active trachoma. The model fitness was assessed using Hosmer -Lemeshow and Omnibus test. Finally, the adjusted odds ratio (AOR) reported with 95% CI.

## Result

### Socio-demographic characteristics of the study participants

In this study 1292 school-aged children participated; the response rate was 97.87%. Nearly half of the study participants were female (55%) and in the age group 6–10 (53%). The majority of study participants (86%) were rural. Moreover, 46% and 12% of the study participants were from 1<sup>st</sup> cycle and 2<sup>nd</sup> cycle primary schools, respectively. Also, 42% of the participants did not enroll in school (Table 1).

### Environmental characteristics of the households and surrounding

In this study, (55%) and (56%) of the HH have improved water sources and sanitation. Only 17% of the HH get water within premises. The latrine coverage was 572(44%), but 729(56%) of the child defecate in the field. In 819(63%) and 468 (36%) of the compound of the HH and around the HH human feces and animal feces were observed respectively. In 503(39%) of the HH domestic animal were present. In 956(74%) of the HH the waste disposal system was improper, and in 567 (44%) of the HH flies were observed (Table 2).

### Behavioral characteristics of the school aged children

In this study only 374(29%) of school aged children wash their hands and face daily. But only 6% of them uses soap to wash their hands and face. Also, nasal and ocular discharge were observed in 203 (16%) and 220 (17%) of school aged children respectively. Half 646 (50%) of school aged children participated in the study lack facial cleans (Table 3).

**Table 1. Socio-demographic characteristics of study participant (n = 1292).**

| Variables Category | | Frequency(n) | Percent (%) |
|---|---|---|---|
| Place of residence | Urban | 155 | 14 |
| | Rural | 1087 | 86 |
| Number children aged 1–9 | One | 653 | 51 |
| | Two or more | 639 | 49 |
| Family size | 1–2 | 438 | 34 |
| | 3–5 | 439 | 34 |
| | Above 5 | 415 | 32 |
| Wealth index | Poor | 609 | 47 |
| | Medium | 420 | 33 |
| | Rich | 263 | 20 |
| Sex of the child | Male | 582 | 45 |
| | Female | 710 | 55 |
| Age of the child | 6–10 | 680 | 53 |
| | 11–16 | 612 | 47 |
| Child enrolment status | Enrolled | 747 | 58 |
| | Not enrolled | 545 | 42 |
| Child educational level | Not enrolled | 545 | 42 |
| | First cycle | 594 | 46 |
| | Second cycle | 153 | 12 |

HH: Household

**Table 2. Environmental characteristics of the households and surrounding.**

| Variables Category | | Frequency(n) | Percent (%) |
|---|---|---|---|
| Source water for HH | Unimproved | 575 | 45 |
| | Improved | 717 | 55 |
| Time to fetch water | 1–30 minute | 214 | 17 |
| | 30–60 min | 740 | 57 |
| | Above 60 min | 338 | 26 |
| Availability of latrine facilities at HH | Yes | 568 | 44 |
| | No | 724 | 56 |
| Latrine type | Improved | 572 | 44 |
| | Not improved | 720 | 56 |
| Utilization of toilet by the child | Latrine | 563 | 44 |
| | Open field | 729 | 56 |
| Feces near the HH | Yes | 819 | 63 |
| | No | 473 | 37 |
| Domestic animal in the HH | Yes | 503 | 39 |
| | No | 789 | 61 |
| Fece of animal around HH | Yes | 468 | 36 |
| | No | 824 | 64 |
| Is there a flies seen at HH | Yes | 567 | 44 |
| | No | 725 | 56 |
| Waste disposal system | Proper | 336 | 26 |
| | Improper | 956 | 74 |

HH: Household

**Table 3. Behavioural characteristics of school-aged child, Southwest Ethiopia, 2021.**

| Variable's Category | | Frequency(n) | Percent (%) |
|---|---|---|---|
| Face washing habit | Yes | 374 | 29 |
| | No | 918 | 71 |
| Frequency face washing daily | Do not wash | 918 | 71 |
| | Once a day | 170 | 13 |
| | Two or more | 204 | 16 |
| Using soap to wash hands and face | Water only | 297 | 23 |
| | Soap with water | 76 | 6 |
| | Do not wash | 919 | 71 |
| Clean nail | Yes | 241 | 19 |
| | No | 1051 | 81 |
| Nasal discharge | Yes | 203 | 16 |
| | No | 1089 | 84 |
| Ocular discharge | Yes | 220 | 17 |
| | No | 1072 | 83 |
| Flies y on the child face | Yes | 203 | 16 |
| | No | 1089 | 84 |
| child's face | Clean | 646 | 50 |
| | Unclean | 646 | 50 |

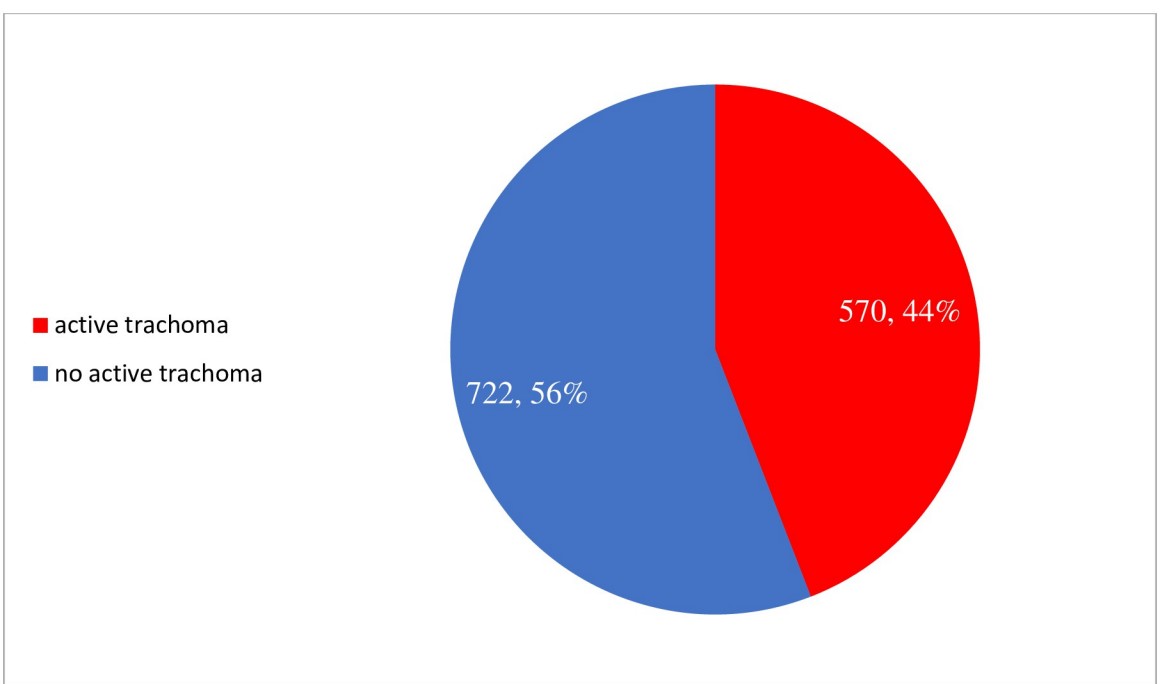

**Fig 1. Prevalence of trachoma among school-aged children in Southwest Ethiopia Peoples Regional States, 2021.**

## Prevalence of active trachoma

In this study, the overall prevalence of active trachoma among school aged children in was 44.12 with 95% CI (41.4, 46.9) (Fig 1).

## Factor associated with active trachoma

In bivariable logistic regression age group, gender, number of children aged 1–9 years, water source, face washing, availability of latrine, sanitation, human and animal feces at HH, flies in the HH, ocular and nasal discharge, flies on the child's face, and condition of child's face were variables showed association at p value less than 0.2. In the multiple logistic regression, age group, gender, water source, sanitation status, flies at HH, ocular discharge, nasal discharge, flies on the child's face, and condition of child's face were the variables significantly associated with active trachoma at P value less than 0.05.

Accordingly, the odds of active trachoma were 2.15 times higher among study participants in the age group 6–10 years than those in the age group 11–16 year [adjusted OR = 2.15 (95% CI: 1.51–3.06)]. Similarly, the odds of active trachoma were 15.36 times higher among female children than males [adjusted OR = 15.36 (95% CI: 10.79, 21.86)]. Also, the odds of active trachoma were decreased by 53% and 31% among study participants who have access to improved water source and improved sanitation at their HH [adjusted OR = .47 (95% CI: (.33,.67)]and [adjusted OR = .31 (95% CI: (.22,.59)] respectively.

In addition, the odds of trachoma were 16.26 and 7.19 times higher among study participant in whom HH flies were observed at their HH and on their face [adjusted OR = 16.26 (95% CI: (9.93,26.64)] and [adjusted OR = 7.19 (95% CI: (3.87,13.34)] respectively. On top of this, the odds of active trachoma were 10.18 and 1.61 times higher among study participants

**Table 4.  Bivariable and Multiple variable analysis of factors associated with trachoma among School aged children in Southwest Ethiopia, 2021.**

| | | Trachoma | | COR (95% CI) | AOR (95% CI) |
|---|---|---|---|---|---|
| | | Yes | No | | |
| Age of the child | 6–10 | 253 | 427 | 1.81(1.45,2.27) * | 2.15(1.51,3.06) ** |
| | 11–16 | 317 | 295 | 1 | 1 |
| Sex of the child | Male | 435 | 147 | 1 | 1 |
| | Female | 135 | 575 | 12.60(9.67,16.4) * | 15.36(10.79,21.86) ** |
| Number of Children in HH | One | 307 | 346 | 1 | 1 |
| | Two or more | 263 | 376 | 1.27(1.01,1.58) * | 1.10(.79,1.53) |
| Water source | Unimproved | 293 | 424 | 1 | 1 |
| | Improved | 277 | 298 | .71 (.59, .92) * | .47(.33,.67) ** |
| Face washing | Yes | 87 | 287 | 3.66(2.79, 4.81) * | 6.56(4.24,10.16) |
| | No | 483 | 435 | 1 | 1 |
| Availability of latrine | Yes | 305 | 263 | 1 | 1 |
| | No | 265 | 459 | 2.01(1.60,2.51) * | 2.12(.13,34.09) |
| Sanitation status | Improved | 308 | 264 | .49(.39, .61) * | .31(.02, 4.59) ** |
| | unimproved | 262 | 458 | 1 | 1 |
| Feces near the HH | Yes | 321 | 498 | 1.3(1.37, 2.17) * | 2.42(.97,6.06) |
| | No | 249 | 224 | 1 | 1 |
| Fece of animal around HH | Yes | 195 | 273 | 1.17(.93,1.47) * | .75(.48,1.18) |
| | No | 375 | 449 | 1 | 1 |
| Flies seen at HH | Yes | 298 | 269 | 1.85(1.47,2.30) * | 16.26(9.93,26.64) ** |
| | No | 272 | 453 | 1 | 1 |
| Nasal discharge | Yes | 114 | 89 | 1.77(1.31, 2.40) * | 1.61(1.06, 2.46) ** |
| | No | 456 | 633 | 1 | 1 |
| Ocular discharge | Yes | 36 | 184 | 5.07(3.48, 7.39) * | 10.18(5.84,17.76) ** |
| | No | 534 | 538 | 1 | 1 |
| Flies on child face | Yes | 54 | 149 | 2.49(1.78, 3.46) * | 7.19(3.87,13.34) ** |
| | No | 516 | 573 | 1 | 1 |
| Child's face | Clean | 312 | 334 | 1 | 1 |
| | Unclean | 258 | 388 | 1.41(1.13, 1.75) * | 2.29(1.57,3.34) ** |

* p value < 0.2,

** p value < 0.05

who have ocular discharge and nasal discharge [adjusted OR = 10.18 (95% CI: (5.84,17.76)] and [adjusted OR = 1.61 (95% CI: (1.06, 2.46)] respectively.

Moreover, the odds of active trachoma were 2.29 times higher among study participants whose faces were unclean [adjusted OR = 2.29 (95% CI: 1.57, 3.34)] (Table 4).

## Discussion

In this study the prevalence of active trachoma among school-aged children was very high 44.1%. This result is in line with findings from adjacent regions and national report [12,31,32]. However, the finding is very high when compared with similar studies in Ethiopia and abroad [13,16,17,19,33–37]. The geography, environment, and population variation can be the reason the result difference. The current study assessed children age ranges 6–16 year; while the earlier research assessed pre-school-aged children [19,33], children aged 1–8 year [17], children aged 1–9 year [13,16,35,36] and children aged 5–9 year [37]. Beyond this, the earlier research

had been carried out prior to the COVID-19 pandemic; yet, the pandemic had impeded the progress of the 2020 trachoma elimination goal [38].

In this study, the odds of active trachoma were 2.15 times higher among children in the age group 6–10 year as compared to those in the age group 11–16 year. This result is in line with finding from similar research [37,39]. Its due to the fact that younger children do not follow hygiene practice to protect them self from trachoma [2,39,40].

Also, the odds of active trachoma were 15.36 times higher among female school-aged children than male. The finding is supported by similar research findings and other evidence [13,39,41]. The gender disparity in acquiring active trachoma among girls can be associated with the gender role that baby girls play in taking care of younger family members who are potential sources of trachoma infection. This disproportionate risk can persist even in the later age of the girl as a woman because of the raising of children, which increases the recurrent acquisition of the infection, complications, and blindness.

In addition, the odds of active trachoma decreased by 53% and 69% among school-aged children who have access to improved water sources and improved sanitation in their HH. This is in line with the study findings, because lack of access to improved WASH facilities in HH is a barrier to maintain personal and environmental hygiene [16,33,36]. Majority of HH in Ethiopia lacks access to improved WASH, children living in this HH are at increased risk of trachoma [26].

Subsequently, the odds of active trachoma were 16.26 and 7.19 times higher among participants in whom HH and face flies were observed, respectively. The finding was in line with results from similar studies because certain species of flies act as mechanical vectors for trachoma infection by transporting the bacteria from infected children to healthy children [2,16,21,37,42].

Moreover, the odds of active trachoma were 10.18, 1.61, and 2 times higher among school-aged children on whom ocular and nasal discharge were observed and whose faces conditions were unclean, respectively. The result is in line with findings from similar studies [12,13,16,19,33]. This is because an unclean face, nasal discharge, and ocular discharge attract flies, which transmit the bacteria from infected children to healthy children [2].

## Strengths and limitations of the study

The limitation of this study was cross-sectional nature of the design, which is difficult to show temporal relation between variables. Social desirability bias was also potential limitation, but using observation check list had minimized it. Moreover, addressing most trachoma impacted population, with relatively large sample size and large geographical coverage is the strength of the study.

## Conclusion

This study found that the prevalence of active trachoma among school-aged children was very high when compared with the WHO threshold. Also, age group 6–10, female gender, presence of ocular and nasal discharge, unclean face of the child's, presence of flies on the child's face, and at HH have significantly associated with increased odds of active trachoma among school-aged children, while access to improved water sources and improved sanitation lowered the odds of active trachoma among school-aged children. Thus, to reduce the high prevalence of active trachoma in the study area, intensifying and scaling up the facial cleanness and environmental and sanitation components of the SAFE strategy is needed by giving great attention to those in the age group 6–10 and female gender.

## Supporting information

**S1 File. Questionnaire.**
(DOCX)

**S2 File. Trachoma data set.**
(DOCX)

**S3 File. Observation check list.**
(CSV)

## Acknowledgments

We would like to acknowledge the zonal health departments and district health offices for their cooperation. Also, our appreciation goes to the guardians of the children who participated in this study for their time and valuable responses. Finally, we would like to thank data collectors and supervisors for their dedication.

## Author Contributions

**Conceptualization:** Dawit Getachew, Fekede Woldekidan.

**Data curation:** Dawit Getachew, Fekede Woldekidan, Gizachew Ayele, Teshale Worku, Mengistu Ayenew, Biruk Bogale, Abyot Asres.

**Formal analysis:** Dawit Getachew, Mengistu Ayenew, Abyot Asres.

**Funding acquisition:** Dawit Getachew, Fekede Woldekidan.

**Investigation:** Dawit Getachew, Gizachew Ayele, Eyob Tekalgn, Teshale Worku, Mengistu Ayenew, Biruk Bogale, Abyot Asres.

**Methodology:** Dawit Getachew, Yordanos Bekele, Samrawit Sleshi, Eyob Tekalgn, Teshale Worku.

**Project administration:** Gizachew Ayele, Mengistu Ayenew, Biruk Bogale, Abyot Asres.

**Software:** Dawit Getachew.

**Supervision:** Dawit Getachew.

**Validation:** Dawit Getachew, Eyob Tekalgn, Teshale Worku, Mengistu Ayenew, Abyot Asres.

**Visualization:** Dawit Getachew, Eyob Tekalgn, Teshale Worku, Mengistu Ayenew, Abyot Asres.

**Writing – original draft:** Dawit Getachew, Mengistu Ayenew, Abyot Asres.

**Writing – review & editing:** Dawit Getachew, Fekede Woldekidan, Gizachew Ayele, Yordanos Bekele, Samrawit Sleshi, Eyob Tekalgn, Teshale Worku, Mengistu Ayenew, Biruk Bogale, Abyot Asres.

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
