## [Decision Letter · Decision Letter 0]

8 Jul 2023

Dear Mr Getachew,

Thank you very much for submitting your manuscript "Magnitude and factors associated with Active Trachoma among Pre-school and School-Aged children in Southwest of Ethiopia" for consideration at PLOS Neglected Tropical Diseases. As with all papers reviewed by the journal, your manuscript was reviewed by members of the editorial board and by several independent reviewers. In light of the reviews (below this email), we would like to invite the resubmission of a significantly-revised version that takes into account the reviewers' comments. 

We cannot make any decision about publication until we have seen the revised manuscript and your response to the reviewers' comments. Your revised manuscript is also likely to be sent to reviewers for further evaluation.

Sincerely,

Joseph M. Vinetz

Section Editor

Joseph Vinetz

Section Editor

Reviewer's Responses to Questions

**Key Review Criteria Required for Acceptance?**

**Methods**

-Are the objectives of the study clearly articulated with a clear testable hypothesis stated?

-Is the study design appropriate to address the stated objectives?

-Is the population clearly described and appropriate for the hypothesis being tested?

-Is the sample size sufficient to ensure adequate power to address the hypothesis being tested?

-Were correct statistical analysis used to support conclusions?

-Are there concerns about ethical or regulatory requirements being met?

Reviewer #1: Objectives not clearly stated. the sampling procedures not clear. e.g., from the study region it's difficult to know the number of kebeles, to have a picture that they were representative. Looks like the authors used multi-stage sampling and would be nice to say from zones how the Woredas an kebeles were selected because the zone I do believe don't have equal Woredas/Kebeles. this would also enable proper description of study population and ensure bias is eliminated through stratification. the sample size is difficult to say weather is adequate because of limited information.

confidentiality of the data, being safe is not well stated.

Reviewer #2: The objectives of the study are mentioned and the study design is appropriate. However the term prevalence would be more appropriate than magnitude in the objective. The title includes preschool children whereas the study was on school children only. This needs correction. The study population is described and appropriate. Sample size is adequate. Ethical considerations should include informed written consent ( not just written consent) and should mention the amount of time permitted from administering the consent by researchers to provision of consent by the participant. The entire manuscript needs copy editing for language and grammar for better understanding

Reviewer #3: The language is confusing and hard to follow.

Proofreading is mandatory.

Validation of the used questionnaire in trachoma?

Was it used in the native language?

**Results**

-Does the analysis presented match the analysis plan?

-Are the results clearly and completely presented?

-Are the figures (Tables, Images) of sufficient quality for clarity?

Reviewer #1: the results are clearly presented.

Reviewer #2: There has been data collected which has not been included in the analysis in the study for example socio-demographics such as marital state, income, educational status, etc, and even other observational data such as cleanliness of nails etc. All this data is irrelevant as it is not a part of the analysis. 

The results are not clear. The figures stated in the table do not match the figures in the manuscript, for example line 306 and 307 in the manuscript says that the the odds for trachoma was 2.35 times higher among children from houses having hand washing facility, whereas in table 2 the odds ratio is 1.2 for those children from houses wit NO handwashing facility. Please reconcile.

Similarly the statement in Discussion line 298,299 and 300 is contradictory to the results in table 2.

Reviewer #3: Review tables and figure labels and numbers

the word "SALES" in the third fig?

**Conclusions**

-Are the conclusions supported by the data presented?

-Are the limitations of analysis clearly described?

-Do the authors discuss how these data can be helpful to advance our understanding of the topic under study?

-Is public health relevance addressed?

Reviewer #1: the limitations not well articulated. the authors poorly discuss how the data can be helpful to advance our understanding of the topic.

the topic is very relevant in public health, but little is addressed.

Reviewer #2: Conclusions are contrary to the data presented as mentioned above. Recommendations are vague and not specific. This study is identical in design to another study carried out in a different area of Ethiopia in preschool children with similar findings and relevant conclusions. ( Reference no 2) What would be interesting however is to discuss the reasons for the significant difference in prevalence in both these studies which has not been done. Other than this difference and its explanation this study has not furthered our understanding of trachoma and its determinants.

Reviewer #3: in the discussion section, it would be catching to the readers' eyes to summarize the reports [23, 28, 39-44], [45, 46] in a table.

**Editorial and Data Presentation Modifications?**

Reviewer #1: Line 266-268: And the odds of trachoma 267 infection was 2.23 times higher among those children who wash their hand after toilet 268 [AOR = 2.23(95% CI: 1.09,2.56)]. This contradicts line 273-274. Authors to relook at the data to ensure its not contradicting what the data says.

Reviewer #2: The entire manuscript needs a good copy editing for better accuracy brevity and clarity. The figures need proper titles. The pie chart for prevalence of trachoma has a title 'Sales'.

Reviewer #3: (No Response)

**Summary and General Comments**

Reviewer #1: Dear Editor,

Thanks for giving me the opportunity to review this paper. The paper addresses an import topic for sub-saharan Africa where Trachoma is endemic yet can be controlled. The Authors have tried to bring out the factors associated with active trachoma among school-aged children. The factors they present are very well know fact the only addition is that this information comes from a new area where such information has not been collected. I do believe they would make this study strong if they looked at the factors of affecting the trachoma elimination in the area as they have alluded to in line 294. 

The objective is not well stated in the article apart from the abstract and there is lot of grammar that need to be corrected to make some statements clear to the reader. The use of abbreviation is common and not expounded e.g., HH could mean hand hygiene. 

 I would recommend the authors to relook at their data analysis and presentation to really make a n impact of the data collected. They seem to be mixing those with higher odds of getting the trachoma.

Authors need to also make the reader have a clear picture of the area of study, by explaining the various division of the areas to general acceptable divisions like towns or villages or clusters. For example, its hand to understand what Woreda and Kebeles means. Are they equivalent of a village or a town or division/district.

In the discussion authors also need to bring out the factors they found as associated with trachoma how it’s different with other published work elsewhere or similar and how does that help with interventions being put in place in Ethiopia.

Reviewer #2: Design of the study is good, however lot of extraneous data has been collected which does not form a part of the analysis and is not discussed and hence irrelevant and also unethical. There is no novelty value and hence the significance, unless it can be shown in the discussion of this study, how it is different from previous ones carried out in Ethiopia on this subject. The general execution has been satisfactorily rigorous but the scholarship leaves a lot to be desired.

Reviewer #3: (No Response)

PLOS authors have the option to publish the peer review history of their article (what does this mean?). If published, this will include your full peer review and any attached files.

Reviewer #1: No

Reviewer #2: Yes: Professor Dr Cynthia Arunachalam

Reviewer #3: Yes: Taher Eleiwa
---

## [Decision Letter · Decision Letter 1]

25 Oct 2023

Dear Mr Getachew,

Thank you very much for submitting your manuscript "High prevalence of active trachoma and associated factors among school-aged children in Southwest Ethiopia" for consideration at PLOS Neglected Tropical Diseases. As with all papers reviewed by the journal, your manuscript was reviewed by members of the editorial board and by several independent reviewers. The reviewers appreciated the attention to an important topic. Based on the reviews, we are likely to accept this manuscript for publication, providing that you modify the manuscript according to the review recommendations. 

Sincerely,

Joseph M. Vinetz

Section Editor

Joseph Vinetz

Section Editor

Reviewer's Responses to Questions

**Key Review Criteria Required for Acceptance?**

**Methods**

-Are the objectives of the study clearly articulated with a clear testable hypothesis stated?

-Is the study design appropriate to address the stated objectives?

-Is the population clearly described and appropriate for the hypothesis being tested?

-Is the sample size sufficient to ensure adequate power to address the hypothesis being tested?

-Were correct statistical analysis used to support conclusions?

-Are there concerns about ethical or regulatory requirements being met?

Reviewer #2: Issues mentioned in the previous review have been addressed in the revised manuscript. However it will be ethical to mention in the manuscript, that the data collected in this study is a part of a larger database which was collected to assess the magnitude of NTDs in the region

Reviewer #3: (No Response)

**Results**

-Does the analysis presented match the analysis plan?

-Are the results clearly and completely presented?

-Are the figures (Tables, Images) of sufficient quality for clarity?

Reviewer #2: Issues mentioned in the previous review have largely been addressed. However there are some glaring errors. For example the conclusion states that improved water source and improved sanitation is associated with increased prevalence of active trachoma, whereas in the same conclusion it is stated that presence of animal feces near the HH is associated with increased active trachoma. there is need to use appropriate language or else the meaning conveyed is exactly the opposite.

Reviewer #3: (No Response)

**Conclusions**

-Are the conclusions supported by the data presented?

-Are the limitations of analysis clearly described?

-Do the authors discuss how these data can be helpful to advance our understanding of the topic under study?

-Is public health relevance addressed?

Reviewer #2: Conclusions need to be rewritten keeping in mind the meaning to be conveyed. The likely reasons for the gender disparity needs to explained better. Limitations need to be rewritten for better clarity of the points raised.

Reviewer #3: (No Response)

**Editorial and Data Presentation Modifications?**

Reviewer #2: The whole manuscript needs a proper copy editing for language and grammar. For example: in the introduction part of the abstract, the second sentence begins with a 'Which'. There are spelling mistakes such as bing for being etc,

Reviewer #3: (No Response)

**Summary and General Comments**

Reviewer #2: The issues raised in the previous review have been addressed satisfactorily, except for the language and grammar editing.

Reviewer #3: The authors did well in the revision, however few concerns are raised.

1- Several typos and grammatical errors, for example, the 2nd sentence in the conclusion (same one in the authors summary doc), the word "cheek list", "Inanition"....etc.

2- Regarding the validation, the authors' response is not identical the the text cited. Also, this is not the ideal validation. What was the result of this pilot testing; not mentioned in the results.

PLOS authors have the option to publish the peer review history of their article (what does this mean?). If published, this will include your full peer review and any attached files.

Reviewer #2: Yes: Dr Cynthia Arunahalam

Reviewer #3: No

Figure Files:

Data Requirements:

Reproducibility:

References

---

## [Editor Report · Decision Letter 2]

5 Dec 2023

Dear Mr Getachew,

We are pleased to inform you that your manuscript 'High prevalence of active trachoma and associated factors among school-aged children in Southwest Ethiopia' has been provisionally accepted for publication in PLOS Neglected Tropical Diseases.

Best regards,

Joseph M. Vinetz

Section Editor

Joseph Vinetz

Section Editor

---

## [Editor Report · Acceptance letter]

11 Dec 2023

Dear Mr Getachew,

We are delighted to inform you that your manuscript, "High prevalence of active trachoma and associated factors among school-aged children in Southwest Ethiopia," has been formally accepted for publication in PLOS Neglected Tropical Diseases.

Best regards,

Shaden Kamhawi

co-Editor-in-Chief

Paul Brindley

co-Editor-in-Chief
